# A Comprehensive Evaluation of Deep Learning Models on Knee MRIs for the Diagnosis and Classification of Meniscal Tears: A Systematic Review and Meta-Analysis

**DOI:** 10.3390/diagnostics14111090

**Published:** 2024-05-24

**Authors:** Alexei Botnari, Manuella Kadar, Jenel Marian Patrascu

**Affiliations:** 1Department of Orthopedics, Faculty of Medicine, “Victor Babes” University of Medicine and Pharmacy, 300041 Timisoara, Romania; 2Department of Computer Science, Faculty of Informatics and Engineering, “1 Decembrie 1918” University of Alba Iulia, 510009 Alba Iulia, Romania; 3Department of Orthopedics-Traumatology, Faculty of Medicine, “Victor Babes” University of Medicine and Pharmacy, 300041 Timisoara, Romania; jenel.patrascu@umft.ro

**Keywords:** meniscus tear, deep learning model, MRI, diagnosis, classification

## Abstract

Objectives: This study delves into the cutting-edge field of deep learning techniques, particularly deep convolutional neural networks (DCNNs), which have demonstrated unprecedented potential in assisting radiologists and orthopedic surgeons in precisely identifying meniscal tears. This research aims to evaluate the effectiveness of deep learning models in recognizing, localizing, describing, and categorizing meniscal tears in magnetic resonance images (MRIs). Materials and methods: This systematic review was rigorously conducted, strictly following the Preferred Reporting Items for Systematic Reviews and Meta-Analyses (PRISMA) guidelines. Extensive searches were conducted on MEDLINE (PubMed), Web of Science, Cochrane Library, and Google Scholar. All identified articles underwent a comprehensive risk of bias analysis. Predictive performance values were either extracted or calculated for quantitative analysis, including sensitivity and specificity. The meta-analysis was performed for all prediction models that identified the presence and location of meniscus tears. Results: This study’s findings underscore that a range of deep learning models exhibit robust performance in detecting and classifying meniscal tears, in one case surpassing the expertise of musculoskeletal radiologists. Most studies in this review concentrated on identifying tears in the medial or lateral meniscus and even precisely locating tears—whether in the anterior or posterior horn—with exceptional accuracy, as demonstrated by AUC values ranging from 0.83 to 0.94. Conclusions: Based on these findings, deep learning models have showcased significant potential in analyzing knee MR images by learning intricate details within images. They offer precise outcomes across diverse tasks, including segmenting specific anatomical structures and identifying pathological regions. Contributions: This study focused exclusively on DL models for identifying and localizing meniscus tears. It presents a meta-analysis that includes eight studies for detecting the presence of a torn meniscus and a meta-analysis of three studies with low heterogeneity that localize and classify the menisci. Another novelty is the analysis of arthroscopic surgery as ground truth. The quality of the studies was assessed against the CLAIM checklist, and the risk of bias was determined using the QUADAS-2 tool.

## 1. Introduction

Deep learning has emerged as a game-changing technology in medical imaging, particularly in detecting meniscal tears through knee magnetic resonance imaging (MRI) [1,2,3]. Meniscal tears are a common knee injury, often requiring accurate and timely diagnosis to guide appropriate treatment. Meniscal injuries rank as the second most frequently occurring knee injury, affecting approximately 12% to 14% of individuals, with a prevalence of 61 cases per 100,000 people [4]. Notably, a significant proportion of meniscal tears coincide with anterior cruciate ligament (ACL) injuries, with occurrences ranging from 22% to 86%. In the United States, an estimated 10% to 20% of all orthopedic surgeries pertain to meniscal procedures, leading to around 850,000 patients undergoing such surgeries annually [5,6,7]. A tear in the meniscus can be seen on a knee MRI as an abnormal intra-meniscal signal on T2-weighted or proton-density MR images, clearly touching the meniscus surface. This signal unequivocally contacts the meniscal surface, indicating that the tear is visible on a minimum of two consecutive slices or images captured in different planes [8,9,10]. Categorizing these tears follows the classifications used by arthroscopic surgeons, as illustrated in Figure 1 [11].

Vertical tears commonly resulting from trauma may manifest as longitudinal, radial, or oblique. On the other hand, horizontal tears, primarily associated with degeneration, exhibit a different pattern. Complex tears result from a combination of various orientations [11]. Deep learning algorithms, particularly convolutional neural networks (DCNNs), have demonstrated their capability to assist radiologists and orthopedic surgeons in identifying meniscal tears with remarkable accuracy [12,13]. DCNNs differ from other network architectures. The structure’s main components are convolutional, pooling, and fully connected output layers. A convolution is an operation that applies a filter to an input image to characterize the inter-correlation among pixels. Repeatedly using the same filter to the input will produce an activation map, specifically a feature map. A nonlinear activation function (reLU) to enhance learning is added. Pooling layers are used to represent down-sampling feature maps. After extraction, the features can be interpreted and predicted, such as classifying the object type in an image. The first operation is to flatten the 2D feature maps and add a fully connected output layer. Figure 2 depicts the architecture of a DCNN applied to knee MRIs. DCNNs are flexible and scalable structures that allow more efficient learning of features in the training data than other machine learning techniques. DCNN succeeds in image recognition, area of interest (ROI) segmentation, and object classification [14,15]. DL-based algorithms automatically discover complex features and learn how to detect objects or image segments. The training of the DL models used in knee joint image analysis is primarily supervised with labeled data. However, a common approach is transferring learning to speed up the process. Collecting knee MRIs from hospital archives and labeling tasks are challenging and error-prone tasks [16].

In medical imaging, the integration of deep learning (DL) into knee magnetic resonance imaging (MRI) holds significant promise and concurrently poses distinctive challenges [17]. This article meticulously explores the intricacies of employing DL models to detect meniscal tears, a prevalent yet diagnostically intricate condition affecting the knee joint. The manifold challenges stemming from the diverse forms and locations of meniscal tears within the intricate meniscus prove a formidable task even for experienced radiologists [18].

Addressing these challenges requires an extensive dataset for DL model training, encompassing diverse tear types and severities. The application of DCNNs, specialized for image analysis, introduces its complexities, including optimal configuration, extensive iterations, and intricate fine-tuning. Model training demands substantial computing power, with transfer learning emerging as a strategy to bolster performance [19,20,21].

Post-training, validation, and testing become imperative, involving the assessment of model accuracy and generalization on MRI scans beyond the training set. The ultimate objective is to deploy DL models in clinical settings, provide consistent, efficient, early detection of meniscal tears, and complement healthcare professionals’ expertise [20,21,22,23].

While highlighting the advantages of DL models—consistency, efficiency, and early detection—it is crucial to acknowledge their limitations. The reliance on substantial and diverse datasets, coupled with the need for clinical expertise, underscores the nuanced integration of DL tools into the complex landscape of medical diagnostics [24,25,26,27,28,29].

This systematic review introduces novel approaches compared to similar previous studies: (i) focus is exclusively on meniscal tears, and while previous reviews have explored DL models in conjunction with various knee injuries, none have explicitly outlined meniscal lesion classifications [30,31,32]; (ii) the accuracy of the DL model is studied through meta-analyses that previous reviews have not conducted to the best of our knowledge [30,31,32]; (iii) evaluation of whether the presented DL models outperform radiologist interpretations; (iv) evaluation of the DL model’s performance in classifying meniscal tears and detecting their various types, providing an updated comparison with human understanding; and, most notably, (v) introduction of the novelty of assessing DL model performance against arthroscopic surgery as the standard reference. Previous reviews reported that due to the heterogeneity of the included studies, it was impossible to achieve in-depth statistical comparisons of the output mode [30,31]; (vi) compared to previous reviews, this one assesses the quality of the studies, considering the guidelines set forth by the CLAIM checklist [33].

## 2. Materials and Methods

The current systematic review adhered to the recommended reporting guidelines for systematic reviews and meta-analyses, known as the PRISMA guidelines [34]. We limited the search and inclusion of articles to only those that applied deep learning models to diagnose menisci in magnetic resonance imaging. The reasons for such an approach are manifold. Firstly, due to the increasing popularity of deep learning models, especially DCNNs, we intended to analyze the state of the art and the challenges and limitations of the variety of such models applied to investigating menisci diseases. As previously mentioned, the pathology of the menisci is a frequent issue worldwide and in all age groups. On the other hand, MRI is one of the most common methods of investigation. Deep learning models’ structure and operation differ from other machine learning techniques (e.g., support vector machines, decision trees, random forests, and logistic regression). Typically, these machine learning methods manually select and extract features from images for prediction or classification. Unlike traditional ML methods, DL models can automatically learn and extract hierarchical features from raw data, speeding up the process. Various domains, including medical imaging, have widely adopted DL techniques and network-based computation.

Consequently, we focused on the potential of DL models in diagnosing menisci in MRIs, excluding other machine learning techniques, in the scope of the homogeneity of articles that can be further quantitatively and qualitatively assessed. We have evaluated articles that present deep learning techniques and their potential further to advance a new avenue for diagnosis and treatment strategies. We detail these methods, results, and the implications of the findings for future research directions.

### 2.1. Eligibility Criteria

The inclusion criteria for this study were as follows:All selected studies employed a deep learning model to detect menisci;The studies exclusively focused on diagnosing knee pathology using MRI images;The selected papers were primarily intended to identify meniscal tears alone or in conjunction with other knee abnormalities (ACL);We compared the deep learning models’ performance to human interpretation.

We applied the following exclusion criteria:Review articles or meta-analyses;Studies utilizing AI models other than deep learning;Articles are not written in the English language;Articles addressing the diagnosis of injuries in knees or joints other than meniscal tears.

### 2.2. Information Sources

We extensively searched four reputable sources, MEDLINE (PubMed), Web of Science, Cochrane Library, and Google Scholar, and restricted our search to research articles published in English in peer-reviewed journals or conference proceedings. The search encompassed articles published between 1 January 2010, and 1 September 2023.

The publications have been divided into two distinct sets:Papers where the deep learning (DL) model primarily focused on detecting meniscal tears, characterization, or even classification of the lesions;Articles recommended using a deep learning-based assisting tool in combination with a deep learning-based assisting tool for detecting common knee injuries, such as meniscal tears, ACL injuries, and osteoarthritis.

### 2.3. Search: Keywords

Keywords used to identify relevant articles included the following: deep learning AND meniscal AND tear, OR deep learning AND menisci, OR deep learning AND detection AND meniscal tear, OR artificial intelligence AND detection AND meniscal tears.

### 2.4. Study Selection and Data Collection

To ensure rigorousness, two authors (A.B. and M.K.) independently conducted each phase of the review process, including literature search, study selection, and data extraction. The third author (J.P.) resolved disagreements between the previous two authors.

We gathered the following information for each article:The area of application covered by the article includes meniscal localization, meniscal tear detection, characterizing meniscal tears, and external validation of meniscal tear detection;Details include the number of patients, publication year, database source, type of deep learning model used, data acquisition, standard reference, preprocessing methods, model training, and external validation.

### 2.5. Statistical Analysis

The receiver operator characteristic curves (ROCs) were considered an evaluation metric. The ROC curve plots the valid positive rate (TPR) against the false positive rate (FPR), which shows the performance of a classification model for all classification thresholds (Figure 3) [35]. This curve plots two parameters: the true positive rate (TPR) and the false positive rate (FPR). True positive rate (TPR) is a synonym for recall and is therefore defined as follows:(1)$$TPR=\frac{TP}{TP+FN}$$

False positive rate (FPR) is defined as follows:(2)$$FPR=\frac{FP}{FP+TN}$$

**Figure 3 diagnostics-14-01090-f003:**
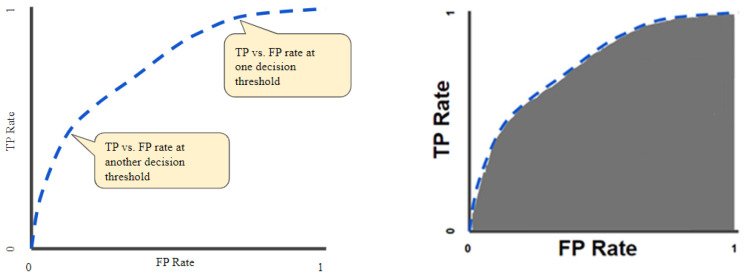
ROC curve (**left**); area under the ROC curve (AUC) (**right**).

The quantitative output of this curve is the area under the ROC curve (AUC), which can be interpreted as an aggregated measure of performance across all possible classification thresholds (Figure 3). The AUC measures the entire two-dimensional area underneath the ROC as a whole curve from (0, 0) to (1, 1).

#### 2.5.1. Meta-Analysis

The meta-analysis compared the pooled quantitative diagnostic accuracy values for the DL models [36,37]. We first retrieved or calculated the sensitivity and specificity values or calculated these values if studies did not provide them [38]. Meta-DiSc, version 1.4, explored heterogeneity with various statistics, including chi-square, I-squared, and Spearman correlation tests and accuracy estimates. We used fixed and random effects models to combine sensitivity, specificity, likelihood ratios, and diagnostic odds ratios for the whole group and different subgroups. Then, based on the collected data, the software produced high-quality figures, including forest plots and summary receiver operating characteristic curves [39].

#### 2.5.2. Quality Assessment

The CLAIM checklist assessed all the included studies’ quality [33]. We extracted and recorded data using standardized forms, specifically a Microsoft Excel spreadsheet. To resolve conflicts over article selection, quality assessment, and data extraction, both observers (A.B. and M.K.) convened a consensus meeting. We scored the items with 0 (not reported), 1 (reported but inadequate), or 2 (registered and adequate). The first group of items (1–13) refers to the quality of the title, abstract, introduction, methods section, study design, and data. The next group (14–24) examines the quality of ground truth, data partitions, and model; items (25–37) refer to the training, evaluation, and results section with data and model performance; and items (38–42) examine the discussion section and other information [33]. Item 41 states that readers can access the full study protocol, which is essential for further investigations and the overall credibility of the study.

#### 2.5.3. Risk of Bias

The QUADAS-2 tool, designed to rate the quality of primary diagnostic accuracy studies [40], was used to look for possible bias in each included study. This tool comprises four principal domains: patient selection, index test, reference standard, and flow of patients through the study, along with the timing of the index test(s) and reference standard (“*flow and timing*”). We evaluated each domain for bias risk and assessed the first three for applicability concerns. We integrated signaling questions to assist in forming a judgment on bias risk. These questions assessed the research methodology’s quality and the study results. BA and PJ thoroughly reviewed all studies to identify bias risks and concerns about applicability. The biased assessment’s findings influenced the data synthesis process by evaluating the relevance and reliability of the generated data [40].

## 3. Results

We conducted a web database search and found 236 articles. After reviewing their title abstracts and applying inclusion and exclusion criteria, we eliminated 199 papers. We then proceeded to read the full text of 37 articles. From these, we excluded papers that introduced AI models other than deep CNNs and those that focused on exclusively segmenting meniscal tears with deep CNNs. Ultimately, our systematic review included 12 articles (Figure 4).

### 3.1. Quality Assessment

The CLAIM-AI checklist assessed the included studies’ specific adherence to the reporting guidelines [33]. Figure 5 illustrates the level of adherence to individual items listed in the checklist from the included studies in the form of a histogram.

All the reported studies (12) clearly stated the study aim, input features, and performance achieved using appropriate metrics. Five studies reported a precise distribution and description of the dataset, and eight clearly described how they established the ground truth (AI’s reference standards). The most prevalent causes of quality point loss were failures to explain ground truth assignments. Lastly, more than half of the studies failed to disclose a conflict-of-interest declaration.

All studies have addressed Items 1–13 very well. Some items have not been discussed in the group between 13 and 25. For example, Items 12, “Describe the methods by which data have been de-identified and how protected health information has been removed”, or 13, “State clearly how missing data were handled, such as replacing them with approximate or predicted values”, and 19, “Describe the sample size and how it was determined”, were not followed by any of the included articles.

Items such as Item 40, “Comply with the clinical trial registration statement from the International Committee of Medical Journal Editors (ICMJE)”, Item 41, “State where readers can access the full study protocol”, Item 42, “Specify the sources of funding and other support and the exact role of the funders in performing the study”, were fulfilled in 50% of the studies or less (Figure 6).

Figure 7 presents the overall situation of the twelve studies and their degree of conformance with the CLAIM checklist.

### 3.2. Risk of Bias

Less than two-thirds of the studies (7 out of 12, 58%) lacked sufficient details to confirm using an appropriate consecutive or random sample of eligible patients. Only half of the studies (6 out of 12; 50%) avoided inappropriate exclusions. Researchers utilized an insufficient sample size in 42% of the included studies (5 out of 12). Consequently, the risk of bias due to participant selection was low in 58% of the studies (7 out of 12) (Figure 8). Concerns about matching the spectrum of participants with the review question requirements were rated as low in 83% of the studies (10 out of 12) (Figure 9). Detailed descriptions of DL models were provided in 83% of the studies (10 out of 12), with almost all studies (11 out of 12; 92%) clearly describing model performance. Features were collected without outcome data knowledge in 83% of the studies (10 out of 12). Thus, the risk of bias related to the index test was low in 92% of the studies (11 out of 12) (Figure 8). Nearly all included studies (11 out of 12; 92%) were judged to have low concerns regarding the alignment of model definition, assessment, and performance with the review question (Figure 9). In 25% of the studies (3 out of 12), the authors did not clearly describe the standard reference for classifying the target condition. In all studies (12 out of 12), researchers assessed the outcome of interest (DL performance) using appropriate tools. In 83% of the studies (10 out of 12), the outcome was consistently defined for all participants without the knowledge of reference standard results. Consequently, the risk of bias due to the reference standard was low in 75% of the studies (9 out of 12) (Figure 8). The reference standards used varied among studies, including those of several radiologist specialists and confirmed arthroscopy of meniscal injuries. Nearly all studies (11 out of 12; 92%) had lows regarding outcome definition, timing, or determination alignment with the review question (Figure 9). In 83% of the studies (10 out of 12), all enrolled participants were included in the data analysis. Data preprocessing was appropriately conducted in 92% of the studies (11 out of 12), and the breakdown of training, validation, and test sets was adequate. Model performance evaluation used suitable measures in 75% of the studies (9 out of 12). Accordingly, 83% of the studies (10 out of 12) had low bias risk in the analysis domain (Figure 8). Reviewers’ judgments about each domain’s risk of bias and applicability concerns for each included study are presented in Figure 10. The QUADAS-2 score assessed the risk of bias by determining whether each included study has low, high, or unclear risks concerning patient selection, index test, reference standard, flow, and timing. Additionally, an analysis of applicability concerns was incorporated.

The assessment results of bias risk within the included studies were conducted using a modified version of QUADAS-2, evaluating four domains: participants, index test, reference standard, and analysis. The term “low risk” (green) indicates the number of studies with minimal bias risk in each domain. “Unclear” (yellow) denotes studies with an uncertain bias risk due to insufficient reported information. “High risk” (red) indicates the number of studies with notable bias risk within the respective domain.

The findings regarding applicability concerns within the included studies were obtained using a modified version of QUADAS-2, examining three domains: participants, index test, and reference standard. “Low risk” (green) indicates the number of studies with minimal concerns regarding applicability within each domain. “Unclear” (yellow) denotes studies with uncertain applicability concerns due to insufficient reported information. “High risk” (red) indicates the number of studies with notable applicability concerns within the respective domain.

### 3.3. Meta-Analysis

When describing accuracy results from several studies, it is essential to determine the magnitude and precision of the accuracy estimates derived from each study and assess the presence or absence of inconsistencies in accuracy estimates across studies (heterogeneity). As accuracy estimates are paired and often interrelated (sensitivity and specificity), it is necessary to report these simultaneously [39]. Exploring heterogeneity is critical to a) understanding the factors influencing accuracy estimates and b) evaluating the appropriateness of statistical pooling of accuracy estimates from various studies. One of the primary causes of heterogeneity in test accuracy studies is the threshold effect, which arises when sensitivities and specificities differ. When a threshold effect exists, there is a negative correlation between sensitivities and specificities (or a positive correlation between sensitivities and 1-specificities), which results in a typical pattern of a “shoulder arm” plot in the SROC space [35].

The validated results for each DL model were used for the meta-analysis. The area under the receiver operating curve (AUC), sensitivity, and specificity for diagnosing meniscus tears were reported separately in the meta-analysis for torn meniscus identification and medial and lateral tear locations, respectively. Multi-class outcomes were converted into binary outcomes. Four studies were excluded from the meta-analysis due to incomplete data [41,42,44,49].

The tear identification analysis articles identified a meniscal lesion in overall knee MRIs. The tear location analysis divided each meniscus into two horns (medial and lateral) and reported the diagnostic performance of the DL model in individual horns [43,45,48].

#### 3.3.1. Meniscal Tear Identification Analysis

Eight studies were included in the tear identification analysis of meniscal tears [12,13,43,45,46,47,48,50]. The dataset was selected based on the Cochrane guidelines for inter-study variation. As the true effect size for all studies was unlikely to be identical due to methodological and interpretation differences, a random effects model was fitted for estimation with partial pooling. Subgroup analyses were conducted to explore the sources of heterogeneity. Once heterogeneity was minimized and outliers were removed, summary estimates were given for the prediction accuracy of AI models. All data analysis and visualization were performed using the Meta-DiSC tool [39].

The pooled sensitivity for the use of DL models was 0.83 (95% CI 0.81–0.86), and the pooled specificity was 0.85 (95% CI 0.84–0.87, Figure 11 left), with an AUC value of 0.915 (Figure 11 right). The heterogeneity was significant for the sensitivity analysis (I-squared = 69.6%), with a statistically significant Cochrane Q statistic of *p* < 0.0017. I-squared was also substantial for the specificity analysis (72.1%) (*p* = 0.007).

Figure 11 shows a forest plot representing the reported sensitivity and specificity values for tear identification analysis. The sensitivities and specificities with a 95% confidence interval of individual studies were indicated by squares and lines extending from their centers. The pooled sensitivities and specificities were displayed in bold and as diamonds in the graphs. In the right part, receiver operating curves (ROCs) for tear identification and the SROC curve indicate the summary estimate in a circle. Circles represent the included study, with dotted lines representing the confidence interval and solid lines for the SROCs. The AUC values are displayed in the legend. SROC = summary receiver operating characteristic.

#### 3.3.2. Medial Meniscal Tear Localization Analyses 

Three studies were included in the medial meniscal tear location analysis [43,45,48]. The pooled sensitivity for the use of DL models was 0.86 (95% CI 0.82–0.90), and the pooled specificity was 0.86 (95% CI 0.82–0.90), with the AUC value of 0.925 (Figure 12). The heterogeneity was low; the I-squared was 8.6% for the sensitivity analysis, and the specificity analysis was zero. 

#### 3.3.3. Lateral Meniscal Tear Localization

Three studies were included in the medial meniscal tear location analysis [43,45,48]. The pooled sensitivity for the use of DL models was 0.64 (95% CI 0.57–0.72), and the pooled specificity was 0.88 (95% CI 0.85–0.91), with an AUC value of 0.844 (Figure 13). The heterogeneity was zero for the sensitivity analysis. I-squared for the specificity analysis was moderate, 38.2%.

### 3.4. Knee MRI Sequence Acquisition

In the examination of various studies, it was observed that five out of the total studies (42%) employed a 1.5 T MRI field to acquire knee scans [12,43,45,48,49]. Additionally, nine studies (75%) utilized a 3 T MRI field for knee scans [12,13,43,44,45,46,47,49,50], while four studies (33%) opted for a combination of both 1.5 T and 3 T MRI for knee examinations [12,43,45,49]. Two studies did not provide information regarding the MRI field used [41,42].

Concerning the image analysis, three studies (25%) included in this review focused solely on sagittal-plane images [42,46,47]. In contrast, only two studies incorporated two-plane images, specifically sagittal and coronal views [45,48]. Moreover, three studies (25%) examined images in all three planes, including sagittal, coronal, and transversal opinions [12,43,49]. Notably, three included studies utilized 3D images for their analysis [13,44,50].

### 3.5. MRI Dataset 

The reporting of dataset composition for creating deep learning models varied across the studies. The included research outlined various datasets, encompassing parameters such as the number of patients, the quantity of MRI images, and the number of MRI scans and slices. Notably, two studies [41,42] exclusively mentioned images. In most studies, 10 out of 12 (83%) disclosed the total number of MRI scans or exams in their datasets. Approximately 50% of the articles provided information on the total number of patients and the total count of knee MRI scans [13,43,46,47,48,50]. Only one study [46] reported comprehensive details, including the total number of patients, the quantity of MRI scans, and the number of slices per examination.

### 3.6. Classification

Out of all the studies analyzed in this review, only one, led by Shin et al. [48], successfully employed a deep learning model to classify meniscal tears, aligning with the commonly utilized MRI and arthroscopic classification systems for such tears [11]. Shin et al. conducted a retrospective study on 1048 knee MR images, including 599 with meniscal tears and 449 without. They developed a DL model, supplemented by a convolutional neural network (CNN) algorithm, to detect tears and classify their types. The DL model achieved AUC values of 0.889 for medial meniscus tears, 0.817 for lateral meniscus tears, and 0.924 for both. For tear classification, AUC values were 0.761 (horizontal), 0.850 (complex), 0.601 (radial), and 0.858 (longitudinal). Data were split into training (70%) and test (30%) sets for evaluation [48].

Other studies concentrated on localizing and detecting meniscal tears within different anatomical regions of the meniscus. As an illustration, Couteaux et al. [41] utilized a Mask R-CNN model to localize menisci and identify tears, achieving both objectives. They trained five models on a full training dataset of 1128 images and five more on an initial batch of 257 images. Performance was evaluated on a test set of 700 images, yielding a score of 0.906, tying for first place with another team in a French competition [41].

Roblot et al. [42] devised an algorithm using fast-region CNN and faster-region CNN, trained on 1123 knee MR images. It addressed three tasks: locating both horns, detecting tears, and determining tear orientation. It scored 0.95 on the training set and achieved AUC values of 0.94 for tear presence, 0.92 for horn position, and 0.83 for tear orientation on the final 700-image dataset. This contributed to an overall score of 0.90 [42]. 

In the study by Fritz et al. [43], a deep convolutional neural network (DCNN) was trained on 18,520 knee MRI studies, with 2000 reserved for validation and testing. Applied to 100 patients undergoing knee MRI and surgery, it detected meniscus tears with AUC values of 0.882 (medial), 0.781 (lateral), and 0.961 (general), with a 95% confidence interval of 0.902 to 0.990 [43].

Tack et al. [44] introduced a technique to identify meniscal tears in specific subregions using 3D MRI data, enhancing precision. Utilizing a 3D convolutional neural network (CNN), they directly analyzed 2399 MRI scans from the Osteoarthritis Initiative Database. Three approaches were explored:Full-Scale Approach: The AUC values for the medial meniscus (MM) ranged from 0.74 to 0.85 and for the lateral meniscus (LM) from 0.91 to 0.94.BB-Crop Approach: Achieved higher AUC values (MM: 0.87 to 0.89; LM: 0.91 to 0.95).BB-Loss Approach: Yielded the highest AUC values (MM: 0.93 to 0.94; LM: 0.91 to 0.96) [44].

Rizk et al. used a 3D CNN to diagnose meniscal tears, training and validating with 8058 examinations. Their integrated architecture, combining meniscal localization and lesion classification, achieved notable results, with AUC values of 0.93 (95% CI: 0.82–0.95) for medial tears, 0.84 (95% CI: 0.78–0.89) for lateral tears, 0.91 (95% CI: 0.87–0.94) for medial tear migration, and 0.95 (95% CI: 0.92–0.97) for lateral tear migration [45].

Li et al. employed Mask R-CNN to diagnose meniscal injuries. They annotated MRI images from 924 patients, dividing the dataset for training, validation, internal testing, and external validation. An additional test dataset from arthroscopic surgery comprised 40 patients. The MRI images were segmented into ten categories. Average precision for menisci categories ranged from 68% to 80%, with accuracies of 87.50%, 86.96%, and 84.78% for healthy, torn, and degenerated menisci, respectively [46].

Li Yz et al. [47] developed a 3D-mask RCNN model for meniscus tear detection and segmentation. Their dataset included 546 knee MRI scans, with 382 for training and 164 for testing. Compared with musculoskeletal radiologists, the model achieved an AUC value of 0.907, accuracy of 0.924, sensitivity of 0.941, and specificity of 0.785 in the test set [47].

Bien et al. [12] developed a CNN MRNET model to diagnose knee injuries, including meniscal tears. They used a dataset of 1370 knee MRI exams, with 1104 abnormal cases (approximately 80.6%), including 508 meniscal tears (about 37.1%). Validation was performed on 917 knee MRI exams. The model achieved specificity, sensitivity, and accuracy (AUC) values of 0.741, 0.710, and 0.725 in detecting meniscal tears [12].

Pedoia et al. [13] evaluated 2D U-NET’s capability in detecting and classifying meniscus and patellofemoral cartilage lesions using 1481 knee MRI exams. They employed the WORMS classification for meniscal tear severity. The CNN showed 81.9% sensitivity, 89.9% specificity, and an 89% AUC for meniscal tear detection. It also achieved an accuracy of 80.74% for intact menisci, 78.02% for mild-to-moderate tears, and 75% for severe tears [13].

Astuto et al. [50] developed a comprehensive automated model for knee assessment using 1435 MRI studies. They categorized severity based on WORMS grades, achieving an AUC of 0.936 (95% CI: 0.90–0.96) for meniscal horn detection. In binary evaluation on the Holdout-Test dataset, sensitivity and specificity were 85%. In a multiclass review, sensitivities for no lesions, tears, or maceration were 85%, 74%, and 85%, respectively [50].

Tsai et al. designed an “Efficiently Layered Network” to identify meniscal tears. They used the same dataset to validate MRNET [8], achieving AUC values of 0.904 and 0.913 for two distinct datasets. The ELNET deep learning model exhibited a sensitivity of 0.86, a specificity of 0.89, and an accuracy of 0.88 when detecting meniscal tears [49].

### 3.7. Comparison between the DL Model and Human Interpretation

#### 3.7.1. DL Model against MSK Radiological Evaluation

Six studies (50%) [12,13,43,45,46,47] compared the performance of deep learning models for meniscus tear detection and radiologist interpretation. All these studies demonstrated impressive outcomes, but only one study showed that the deep learning model surpassed human interpretation, specifically in the study conducted by Li Yz and his team [47]. Li Yz et al. evaluated their 3D-mask RCNN against musculoskeletal radiologists using the Stoller method for meniscus injury assessment. In the DCNN model, the test set AUC, accuracy, sensitivity, and specificity were 0.907, 0.924, 0.941, and 0.785, respectively. Radiological evaluation yielded AUC, precision, sensitivity, and specificity of 0.834, 0.835, 0.889, and 0.754, respectively. The 3D-mask RCNN significantly outperformed radiological evaluation (*p* = 0.0009, bootstrap test) [47].

Fritz et al. [43] compared DCNNs’ performance with that of two radiologists in their study. The DCNN showed 84% sensitivity, 88% specificity, and 86% accuracy for medial meniscus tear detection. Reader 1 achieved 93% sensitivity, 91% specificity, and 92% accuracy, while Reader 2 achieved 96% sensitivity, 86% specificity, and 92% accuracy. There was a significant difference in sensitivity for detecting medial meniscus tears between Reader 2 and the DCNN (*p* = 0.039), but no significant differences in other metrics [43].

In their study, Rizk et al. [45] evaluated the algorithm’s performance on a test set of 299 examinations reviewed by five musculoskeletal specialists and compared it to general radiologists’ reports. During external validation for medial and lateral meniscal tear detection, the models achieved an AUC of 0.83 (with a 95% confidence interval of 0.75 to 0.90) without further training. With fine-tuning, the AUC improved to 0.89 (with a 95% confidence interval of 0.82 to 0.95) [45].

Li et al. [46] demonstrated a robust performance of their model compared to the diagnosis by a board-certified radiologist with 15 years of experience. In the internal testing dataset of 200 patients, only six samples (3%) were unrecognized by the deep learning model. Additionally, the model accurately identified 49 out of 56 healthy samples, 80 out of 92 torn samples, and 39 out of 46 degenerated samples. The diagnostic accuracies for healthy meniscus, torn meniscus, and degenerated meniscus were 87.50%, 86.96%, and 84.78%, respectively [45].

In the study by Bien et al. [12]., the model’s performance in detecting meniscal tears was assessed in terms of specificity, sensitivity, and accuracy (AUC), resulting in values of 0.741, 0.710, and 0.725, respectively. In comparison, radiologists and orthopedic surgeons participating in the study exhibited superior performance, achieving a specificity of 0.882, a sensitivity of 0.820, and an accuracy (AUC) of 0.849 [12].

Pedoia et al. [12] conducted a comparative analysis between three radiologists and a deep learning model. The average agreements for cases with no meniscus lesion, mild–moderate lesion, and severe lesion were 86.27%, 66.48%, and 74.66%, respectively, among the three experts. In contrast, the best-performing deep learning model obtained agreement rates of 80.74%, 78.02%, and 75.00%, respectively [13].

#### 3.7.2. DL Model against Arthroscopic Surgery Diagnosis

Of all the studies included in this review, only three [43,46,47] compared arthroscopic surgery diagnoses. Li Yz et al. [47] evaluated their DL model on 164 patients graded by the Stoller classification who underwent arthroscopic surgery. The accuracy rates for arthroscopic meniscus tears corresponding to injury grades III and II were 0.846 and 0.66, respectively. Thus, the overall accuracy of MRI grading for knee injuries compared to arthroscopic diagnosis results was 0.835, with sensitivity at 0.889 and specificity at 0.754.

Fritz et al. [43] tested their DCNN model on 100 patients who underwent arthroscopic surgery. They achieved sensitivities, specificities, and accuracies ranging between 84% and 92% for detecting medial and lateral meniscus tears, except for the sensitivity for lateral meniscus tear detection, which was considerably lower at 58%.

Li et al. [46] also obtained promising results with arthroscopic surgery. Among 40 patients with diagnoses confirmed by the gold standard, 87.50% (35 out of 40) were correctly diagnosed using their model. Validation with an external dataset indicated that diagnosing torn and intact meniscus tears using 3.0 T MRI images exceeded 80%, with an accuracy of 87.50% compared to diagnoses made through arthroscopic surgery.

The following tables and diagrams summarize each study’s performance (AUC) (Figure 14) in detecting meniscal tears (Table 1 and Table 2).

## 4. Discussion

This comprehensive review highlights the diversity among the research papers it covers. The primary objective of this study was to assess the capability of deep learning in the identification, location, description, and categorization of meniscal tears within MRI images. This investigation’s findings indicate that various deep learning models exhibit strong performance in detecting and categorizing meniscal tears, occasionally surpassing the expertise of musculoskeletal radiologists [47]. In this review, most of the studies focused on detecting tears in either the medial or lateral meniscus [41,42,43,45,46,50], including pinpointing the tear’s location, whether it is in the anterior or posterior horn [41,42,45,46], with excellent performance, as indicated by AUC values ranging from 0.83 to 0.94. Tack et al. [44] extended the deep learning model to locate tears within the body of the medial or lateral meniscus (LM or MM), achieving a high AUC. Li et al. [46] also demonstrated an 84.78% accuracy in diagnosing degenerated meniscus, which is a remarkable result. This study featured the most extensive set of labels, encompassing conditions like complete absence, anterior or posterior horn tears, meniscus body tears, degenerations, and intact regions. This comprehensive labeling system substantially supports medical professionals when interpreting MRI images.

Pedoia et al. [13] introduced a convolutional neural network (CNN) with a three-class model to categorize severity, achieving accuracy rates of 80.74% for intact menisci, 78.02% for mild-to-moderate tears, and 75% for severe tears. Shin et al.’s study [48], the first to develop a deep learning model for classifying meniscal tears based on knee MRI, is particularly notable. When considering an AUC greater than or equal to 0.9 as outstanding, 0.9 to 0.8 as excellent, and 0.8 to 0.7 as acceptable, the model trained using knee MRI data has strong potential for practical application in diagnosing meniscal tears. Regarding the model’s capacity to distinguish between different types of meniscal tears, the AUC values were 0.761 for horizontal tears, 0.850 for complex tears, 0.601 for radial tears, and 0.858 for longitudinal tears. The performance in identifying radial tears was particularly commendable, and the model showed acceptable performance in identifying the other three types of meniscal tears.

Regarding the DL pipeline adopted by the research teams that produced the results included in the studies under this review, one may notice a common understanding of the process, as presented in Figure 15. However, high variability is identified in the methods and techniques applied along each step of the DL pipeline. The analytical process of this review focused on the assessment of the technical and methodological detail and accuracy used in each step of the pipeline in the reviewed studies, namely the description of the dataset, description of the image preprocessing, description of the ground truth determination, explanation of the ROI localization, description of the evaluation applied to DL models and metrics, and description of the applied methods limitations. Table 3 summarizes the approaches for each step in the DL pipeline.

### 4.1. Description of the Dataset

Table 3 discloses the high variability in initial dataset descriptions regarding terminology. Some authors refer to the input dataset as images [41,42], others as patients [46,47], or images and patients [12,43,45,50], scans [44], cases [48], and images and slices [12,49].

It is well known that the output of an MR investigation for a patient is composed of several imaging sequences, such as T1, T2, and PD, each consisting of many slices depending on the machine’s settings. Also, computer vision practice and some best practices explain which sequence is better for a specific investigation type. The evaluation of knee MRI consists of (a) determination of pulse sequence for review, (b) evaluation of T2-weighted images, (c) evaluation of T1-weighted images, (d) evaluation of specialized pulse sequences, and (e) correlation of imaging findings with clinical history and examination findings. The most used sequences of knee MRI are T1-weighted images, T2-weighted images, intermediate-weighted or proton-density-weighted images, fluid-sensitive sequences such as STIR or fat-suppressed T2-weighted, and gradient-echo images. The sequences can be in the coronal, sagittal, or transversal plane.

On the other hand, not all the slices are helpful to be included in the final dataset that will be provided for the DL model. Moreover, an appropriate selection of curated slices requires expert knowledge. Therefore, the applicability of these methods to 3D volumes is unclear since they were not trained on 3D data. The first limitation of the studies relates to the specific dataset created, which contained only two T2-weighted MR images in the sagittal plane for each patient. In contrast, an MRI examination of the knee usually includes around 100 images [42]. Therefore, most studies fail to precisely describe how the images have been selected and which slices have been included in the further preprocessing.

Most studies are conducted on 2D images taken from one plane, e.g., sagittal or a combination of two planes (sagittal and axial) [41,42,43,45]. Only one study used 3D images that contained all three planes (sagittal, axial, and coronal) [44]. Astuto et al. [50], Pedoia et al. [13], and Shin et al. [48] applied 3D models to 2D images. A significant limitation of such methods is that the trained 2D CNNs cannot take whole MRI volumes into account, thus possibly missing significant feature correlations in 3D space. 

Another variable is the provenance and size of the curated dataset. In some cases, images are provided from a single machine or are preprocessed, as in the French challenge [41,42], provided as a collection obtained from various machines and hospitals, or are part of the Initiative database [13,50]. The size of the dataset varies from a couple hundred to ten thousand (see Table 3). Nonetheless, Couteaux et al. had a limited training dataset of only 300 torn menisci [41]. The initial size of the curated dataset is crucial for the success of the model training process. Consequently, they opted to train a relatively shallow convolutional neural network (CNN) with a structure akin to VGG architecture, utilizing 246 samples for the training dataset and 54 for validation [41]. In the study by Fritz et al., the deep learning-based model was only tested on knee MRIs performed at their institution. Therefore, the results of this study apply to knee examinations using one standard knee protocol and an MR scanner.

Meniscal tear detection relies on more than a single sagittal MRI image in a real-world setting; these results cannot be used in clinical practice [45].

### 4.2. Description of the Image Preprocessing

To obtain the curated dataset ready to fit into the model, all the teams, without exception, had intensely preprocessed the original knee MRIs. Table 3 summarizes the preprocessing techniques applied in each study. The most important preprocessing techniques applied in the studies are as follows:Resizing and resampling: The images varied in resolutions and dimensions. Resizing and resampling the images to a consistent size or spacing was essential for ensuring compatibility with deep learning models requiring fixed input dimensions. Dimensionality reduction was achieved by manual cropping [45] and bounding boxes [44] or by deep models such as UNet [13].Intensity normalization and histogram equalization: The MRIs exhibited varying intensity ranges and contrasts due to differences in acquisition protocols and devices. Intensity normalization scaled the pixel values to a standard range, such as [0, 1] or [0, 255], enhancing the contrast and enabling more meaningful image comparisons. A histogram-based intensity standardization algorithm was applied to the images to account for variable pixel intensity scales within the MRI series [12,49]. A representative intensity distribution was learned from the training set exams for each series. Then, the parameters of this distribution were used to adjust the pixel intensities of exams in all datasets (training, tuning, and validation). Under this transformation, pixels with similar values correspond to similar tissue types. After intensity standardization, pixel values were clipped between 0 and 255, the standard range for PNG images [12,49].Image noise and signal abnormalities reduction: MRIs can be affected by various types of noise, such as Gaussian noise, salt-and-pepper noise, or speckle noise. Noise reduction techniques, such as Gaussian filtering, median filtering, or anisotropic diffusion, reduce noise while preserving important image features. The model’s performance is very much dependent on the image quality of the training dataset. One profound influence on the model’s performance is the image nosiness or the artifacts [42]. Signal abnormalities in images are still a challenge [44]. When menisci with tears are to be distinguished from menisci without tears, signal abnormalities are currently regarded as the latter. A fine-grained differentiation between tears and signal abnormalities is a challenge, primarily because of the ambiguous image appearance [44].Data augmentation: Data augmentation was used to increase the size and diversity of a training dataset by creating new instances by applying various transformations to the original data. Most studies have imbalanced sets for the categories to be used for training. Usually, there are fewer images of healthy meniscus than of torn ones. Also, as in the case of the predefined datasets [41,42], those contained only two T2-weighted MR images in the sagittal plane for each patient, whereas an MRI examination of the knee usually includes around 100 images. Moreover, images were preprocessed to have the same matrix and voxel size. Dataset imbalance may explain the inferior overall performances on lateral meniscal tear detection and characterization. A more significant amount of data, including lateral meniscal tears in the training dataset, may further increase model performances laterally [45]. Tsai et al. [49] performed randomized data augmentations on each series, which included translation, horizontal scaling, and minor rotations up to 10 degrees around the center of the volume. For volumes captured in the axial and coronal orientations, they applied an additional random rotation of a multiple of 90 degrees to the volume. Finally, all the images were resized to 256 × 256 before entering the network. Aside from data augmentation, they implemented oversampling to compensate for dataset imbalances [49].

### 4.3. Description of the ROI Localization

Accurate segmentation of regions of interest (ROIs), in our case, the menisci, is one of the most important steps for the DL model’s performance. However, segmentation can be challenging due to overlapping structures, weak boundaries, or similar intensities between the target region and surrounding tissue. Segmentation techniques ranged from simple thresholding to more complex approaches like active contours or deep learning-based methods. Careful selection and application of segmentation techniques have significantly improved the input images’ quality and enhanced subsequent analysis algorithms’ performance. Table 3 summarizes various techniques applied to the identification of ROIs, such as custom localization with manual annotation tools, class activation maps, the bounding box crop approach, semi-automated crops using ITK-SNAP, segmentation with DL models UNet, fast-CNN, bounding box regression, and 2D and 3D bounding boxes.

Most of the presented methods individually detected meniscal tears for all anatomical subregions of the menisci, i.e., the anterior horn, the meniscal body, and the posterior horn.

### 4.4. Description of the Ground Truth Determination

The model’s performance depends on the ground truth quality used to train and validate it. In AI applications, ground truth is the a priori information considered to be accurate based on objective, empirical evidence. Ground truth refers to the data used in the training phase that teach an algorithm how to arrive at a predicted output and is considered the “correct” answer to the prediction problem the tool is learning to solve. This dataset becomes the standard against which developers measure the accuracy of the system’s predictions.

The ground truth for classifying healthy and torn menisci is obtained by several methods in the studies reviewed. The most objective way to establish the ground truth is through arthroscopy evidence. If this is unavailable, the objectivity is set by two or more MSK radiologists, orthopedic surgeons, or data scientists trained by radiologists who classify the images into two or more categories. The images with ambiguity are then eliminated from the dataset. This procedure is time-consuming and challenging; in some cases, the researchers adopt NLP tools to extract the image’s label from the attached description. However, this method has drawbacks as the text is not standardized and, therefore, in some cases, challenging to interpret by AI processors. Roblot et al. combine medical records and radiology reports through natural language processing techniques [42]. They also propose that radiologists should write their reports in a consistent and structured format. For example, each MR image of the knee was annotated with specific diagnoses referenced in the CSV file to help teams be more efficient and work faster on creating the algorithm [42].

In Rizk’s study, the meniscal tear label extraction from radiological reports occurred without surgical correlation but with internal validation on a subset labeled by expert MSK radiologists and external validation [45].

In the context of this review, the predominant approach for labeling MRI images to train DL models across most of the included papers involved relying on the radiological expertise of a musculoskeletal (MSK) radiologist. However, refs. [45,46] took a different approach in their respective studies, opting for arthroscopic surgery as a more precise method for diagnosing meniscal tears. This choice not only offers greater accuracy but also bolsters the robustness of the DL model. Conducting rigorous comparisons between arthroscopic diagnostic outcomes and the dataset used for training and validation is imperative for future research. This step is essential for further enhancing the model’s accuracy.

### 4.5. Description of the Evaluation Applied to DL Models and Metrics

Some studies identified only meniscal injuries; no distinction was made between different types of meniscus tears [46]. Moreover, the analysis did not compare the diagnostic accuracy of sagittal, coronal, and transverse views. In future studies, it will be necessary to check the accuracy of the proposed model for meniscus tears at different positions by including more cases and investigating the influence of varying layer thicknesses on diagnosing meniscus tears.

The validation method also needed to be improved in many cases. Only a few sample diagnoses were confirmed by arthroscopic surgery. In future research, a rigorous comparison of arthroscopic diagnostic results with the data used for training and verification is required to improve the DL models’ accuracy further.

A difficulty in deep learning for medical imaging is curating large datasets containing examples of the various abnormalities that can occur on a given imaging examination to train an accurate classifier. The deep learning model described in [12] was developed and introduced based on MRI data from one large academic institution. While MRNet performed well on the external validation set without additional training (AUC 0.824), the team saw a substantial improvement (AUC 0.911) after training on the external dataset. This finding suggests that achieving optimal model performance may require additional model development using data like the model is likely to see in practice [12]. More research is needed to determine if models trained on larger and multi-institutional datasets can achieve high performance without retraining.

### 4.6. Description of the Applied Methods’ Limitations

All studies have included limitations in their approaches. A standard limitation among many of the methods listed above is their firm reliance on segmentations of the menisci (or at least of bounding boxes), which can be challenging to obtain due to the inhomogeneous appearance of pathological menisci in MRI data as well as insufficient contrast to adjacent tissues. Most of the approaches consider datasets of 2D images. A significant limitation of such methods is that the trained 2D CNNs cannot take whole MRI volumes into account, thus possibly missing significant feature correlations in 3D space.

Couteaux’s study has pointed out one major limitation connected to the small available database. In this case, training a standard classifier from the image resulted in suboptimal performances, as observed in their initial experiments [41]. They chose to localize and segment the menisci and perform the classification within the anterior and posterior menisci. They opted for the Mask R-CNN approach as it can perform both tasks jointly [41].

## 5. Study Limitations

While this review and meta-analysis offer valuable insights into how deep learning (DL) contributes to identifying meniscal tears on MRI scans, some limitations exist. Firstly, the studies included vary in the datasets used, MRI techniques employed, DL models utilized, and the absence of explicit standard references, potentially leading to result and interpretation discrepancies. Moreover, only a few studies compare DL outcomes with arthroscopic surgery, limiting the applicability of findings to clinical contexts. Arthroscopy is the gold standard for non-invasive knee MRI assessments because it directly evaluates structural integrity, tear patterns, and functional characteristics. Additional studies like those by Ying et al. [51] are necessary to integrate arthroscopic insights into MRI diagnostic models for more precise meniscus tear detection. Furthermore, despite the focus on DL model performance regarding meniscal tear localization, description, or classification, the number of studies addressing these aspects remains relatively small, potentially restricting the depth of analysis. Although several similar reviews [30,31,32,52] exist in the literature, the primary limitations remain the heterogeneity and number of included studies, preventing thorough statistical comparisons of output modes. Addressing these limitations in future research endeavors would enhance the comprehensive evaluation of DL’s role in detecting meniscal tears on MRI scans.

## 6. Conclusions

DL models have great potential, and this is the way to continue investigating MRIs for automated diagnosis of knee pathologies, as exemplified in this study for meniscus pathology. Future DL models must strive to enhance the AUC achievements, aiming for a range better than 0.9. This challenge is underscored by the following limitations that need to be addressed:The quality of segmentation, a crucial aspect in most studies, poses a significant challenge due to the inhomogeneous appearance of the pathological menisci in the MRI data and insufficient contrast in additional tissues [53]. It is imperative to develop better preprocessing techniques and algorithms for automated segmentation. A promising avenue is to design an extension of the Segment Anything Model (SAM) for knee MRIs [54].Most of the models operated on 2D slices. Appropriate selection of such slices is time-consuming and requires expert knowledge. Better algorithms that operate on 3D images are required to understand the pathological menisci of the whole anatomical structure, not only one or two localizations.None of the methods, with one exception [44], can individually detect anatomical subregions of the menisci. Research should be directed into this area.To overcome the scarcity of data, transfer learning from other, more advanced fields with more data should be implied in new model configurations.A significant limitation of DL models is their task-specific nature. These models are typically designed and trained for a specific task, and their performance can degrade significantly when applied to new tasks or different types of imaging data. This lack of generality poses a substantial obstacle to the broader application of these models in clinical practice. Therefore, there is a growing demand for universal models in knee image segmentation and classification: models that can be trained once and then applied to a wide range of segmentation and classification tasks. In our opinion, further work needs to be performed to cover a broader structure analysis of knee components in a structured and standardized way before implementing these tools efficiently in clinical practice. Other studies on deep learning algorithms’ interpretability are also needed to support professional confidence and efficient implementation. Still, the active participation of radiologists in building these models and strong partnerships with data scientists are keys to supporting early adoption in clinical routine. To generalize and bring any DL algorithm to a clinical application, such an algorithm would need to be incorporated into a standardized workflow and an accurate end-to-end diagnostic tool would need to be developed.

## Figures and Tables

**Figure 1 diagnostics-14-01090-f001:**
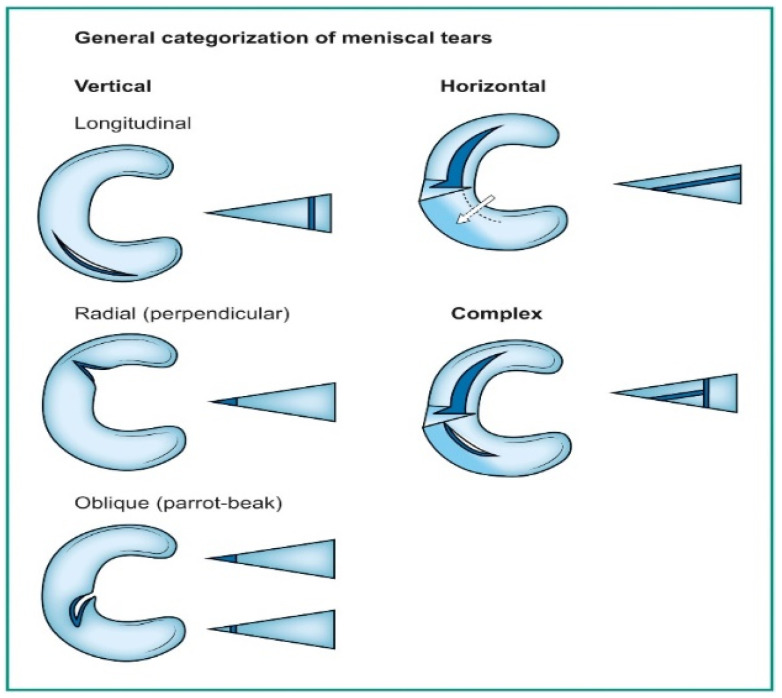
Orientation-based classification of meniscal tears.

**Figure 2 diagnostics-14-01090-f002:**
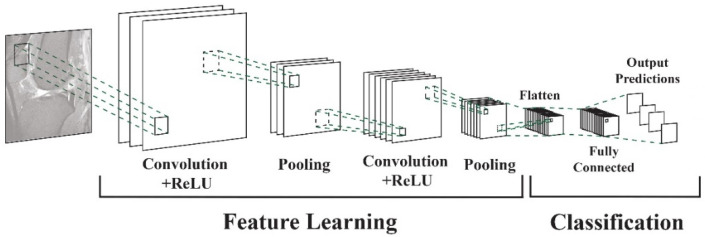
DCNN architecture for knee MRIs [16].

**Figure 4 diagnostics-14-01090-f004:**
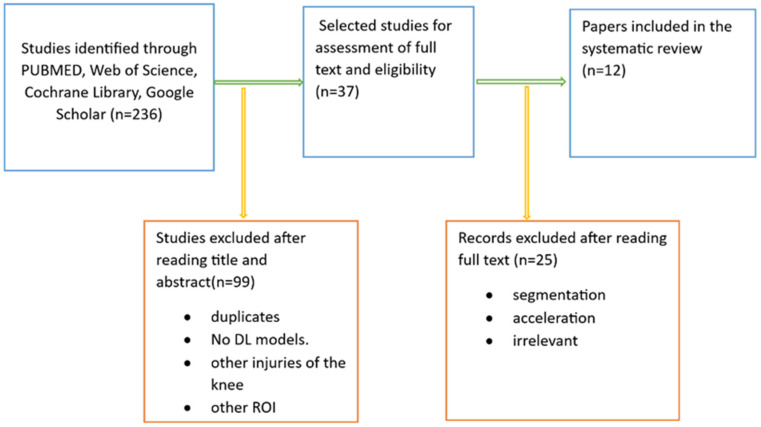
Literature search pipeline.

**Figure 5 diagnostics-14-01090-f005:**
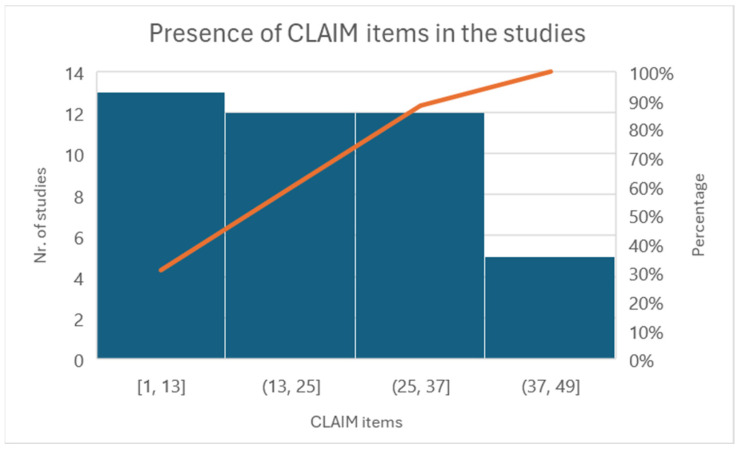
Histogram of individual items of the CLAIM checklist.

**Figure 6 diagnostics-14-01090-f006:**
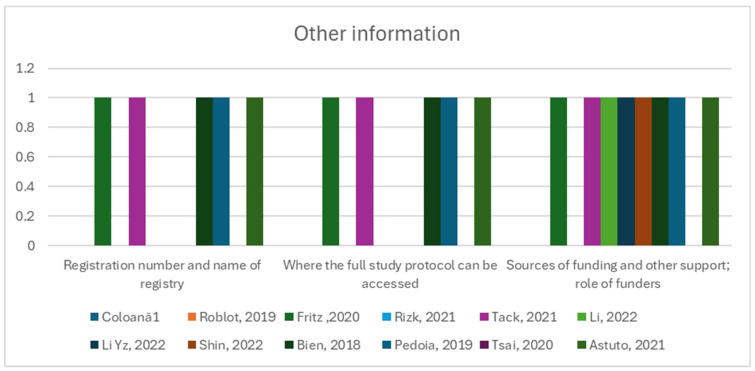
Other information group of items and their observance in the twelve studies [12,13,41,42,43,44,45,46,47,48,49,50].

**Figure 7 diagnostics-14-01090-f007:**
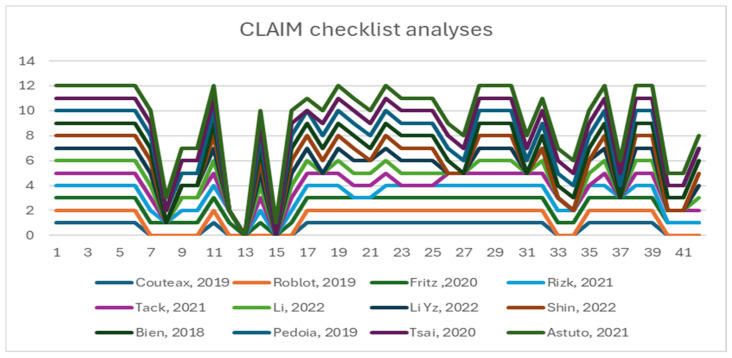
Overall conformance with the CLAIM checklist [12,13,41,42,43,44,45,46,47,48,49,50].

**Figure 8 diagnostics-14-01090-f008:**
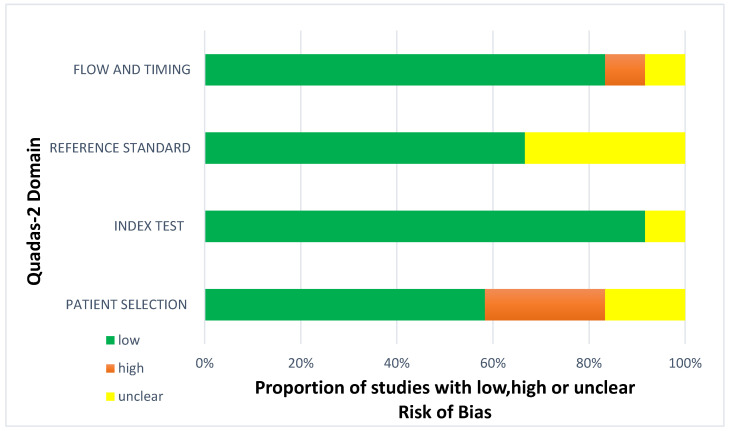
Diagram for risk of bias.

**Figure 9 diagnostics-14-01090-f009:**
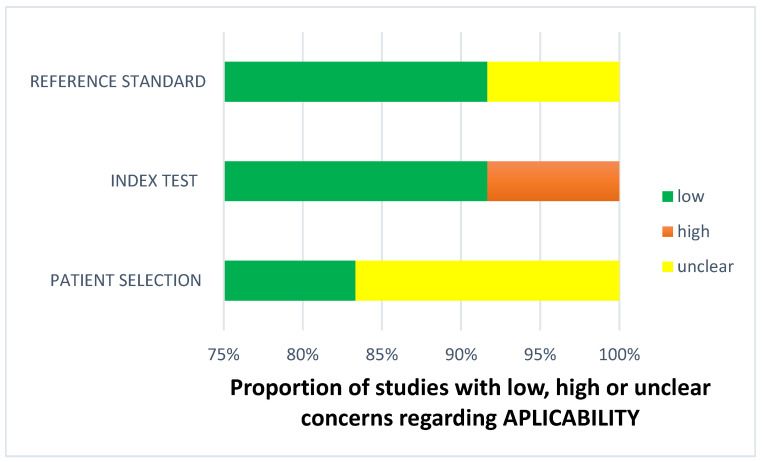
Concerns regarding applicability.

**Figure 10 diagnostics-14-01090-f010:**
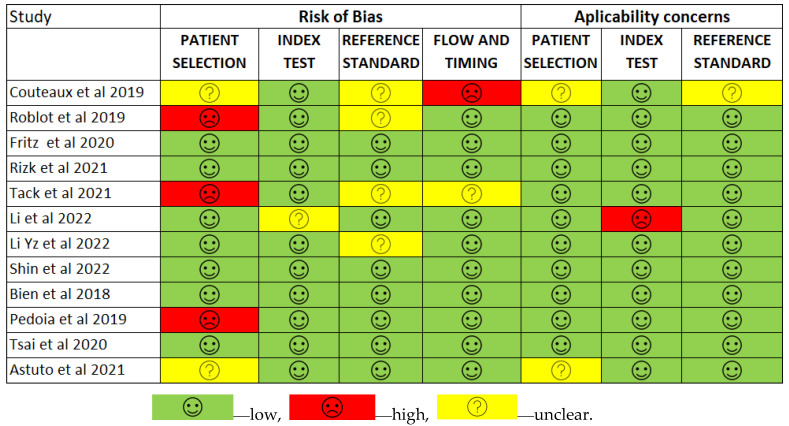
The QUADAS-2 score assesses the risk of bias [12,13,41,42,43,44,45,46,47,48,49,50].

**Figure 11 diagnostics-14-01090-f011:**
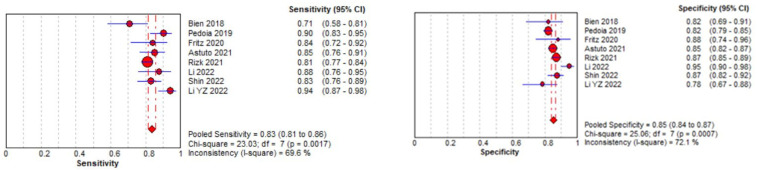
Forest plot representing the reported sensitivity and specificity values for meniscal tear identification analysis. Receiver operating curves (ROCs) for meniscal tear identification analysis and the SROC curve [12,13,43,45,46,47,48,50].

**Figure 12 diagnostics-14-01090-f012:**
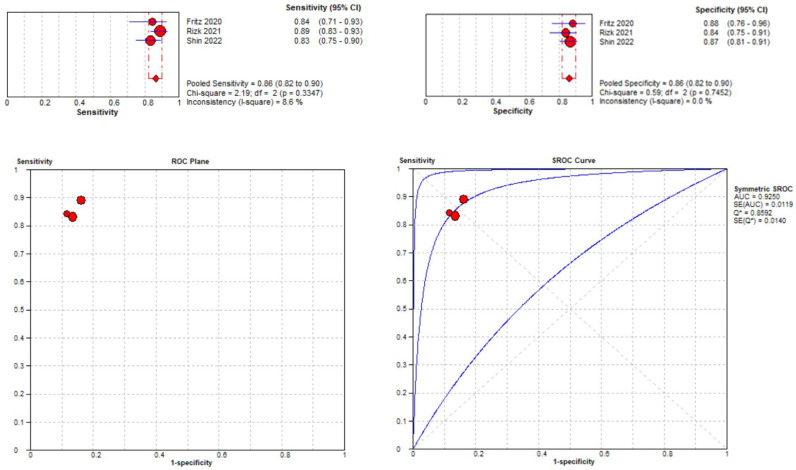
Forest plot representing the reported sensitivity and specificity values for medial meniscal tear identification analysis. Receiver operating curves (ROCs) for medial meniscal tear identification analysis and the SROC curve [43,45,48].

**Figure 13 diagnostics-14-01090-f013:**
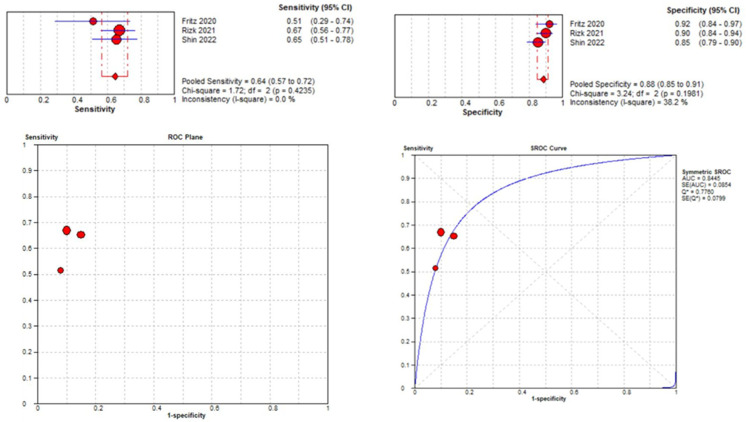
Forest plot representing the reported sensitivity and specificity values for lateral meniscal tear identification analysis, receiver operating curves (ROCs) for lateral meniscal tear identification analysis, and the SROC curve [43,45,48].

**Figure 14 diagnostics-14-01090-f014:**
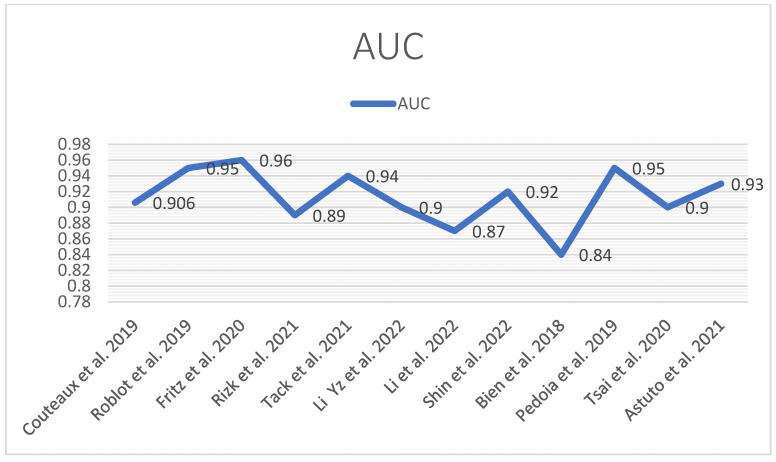
AUC performance diagram [12,13,41,42,43,44,45,46,47,48,49,50].

**Figure 15 diagnostics-14-01090-f015:**
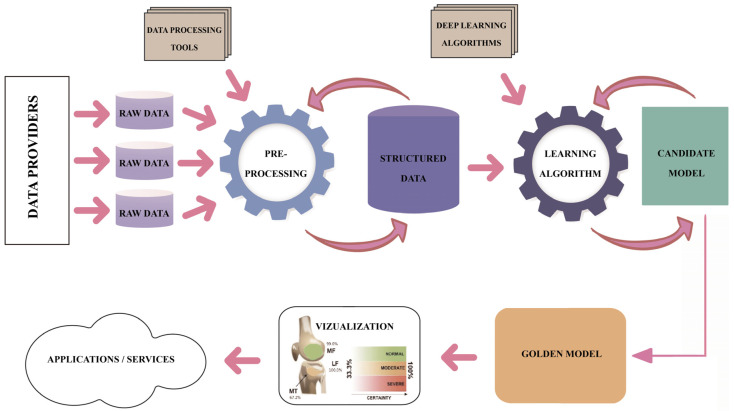
The DL pipeline.

**Table 1 diagnostics-14-01090-t001:** Performance (AUC) of localization and classification of meniscal tears.

	Couteaux et al.	Roblot et al.	Fritz et al.	Rizk et al.	Tack et al.	Li Yz et al.	Li et al.	Shin et al.	Bien et al.	Pedoia et al.	Tsai et al.	Astuto et al.
	[41]	[42]	[43]	[45]	[44]	[47]	[46]	[48]	[12]	[13]	[49]	[50]
Meniscus	0.906	0.95	0.96	0.89	0.94	0.90	0.87	0.92	0.84	0.95	0.90	0.93
Degeneration							0.85					
MM			0.88	0.93	0.94			0.88				
LM			0.78	0.84	0.93			0.81				
AH MM					0.94		0.69					
AH LM					0.96							
PH MM					0.93		0.71					
PH LM					0.91							
Body MM					0.93		0.7					
Body LM					0.91							
Horizontal tear								0.76				
Complex tear								0.75				
Radial tear								0.61				
Longitudinal tear								0.83				

Abbreviation: MM—medial meniscus, LM—lateral meniscus, AH MM—anterior horn medial meniscus AH LM—anterior horn lateral meniscus, PH MM—posterior horn medial meniscus, PH LM—posterior horn lateral meniscus, Body MM—body medial meniscus, Body LM—body lateral meniscus.

**Table 2 diagnostics-14-01090-t002:** Description of the main features of the studies.

Authors	Year	DL Model	MRI (Field/ Sequences)	Task	Metrics (AUC)
Couteaux et al. [41]	2019	MASK R-CNN	N/A	Tear orientation classification: medial and lateral meniscus, anterior or posterior horn, horizontal or vertical tear.	0.906
Roblot et al. [42]	2019	CNN, R-CNN	N/A/ T2 SAG	Tear orientation classification: medial and lateral meniscus, anterior or posterior horn, horizontal or vertical tear.	0.90
Fritz et al. [43]	2020	DCNN	1.5 or 3 T/T1 STIR IWs IW Dixon COR, TRANS, SAG	Tear detection of medial and lateral meniscus	0.961
Rizk et al. [45]	2021	3D CNN	1.0 T 1.5 T 3 T/COR SAG PD FSW	Medial and lateral tear meniscal migration	MM 0.93/ LM 0.84/
Tack et al. [44]	2021	3D CNN	3.0 T/DESS 3D MRI, IW TSE	Detection of lesions: anterior horn, body posterior horn of MM, and LM	MM 0.94/LM 0.93/
Li et al. [46]	2022	Mask R-CNN	3.0 T/ SAG PDW	Detection of normal meniscus, meniscal tear, and degenerated meniscus	0.86 DM 0.84
Li Yz et al. [47]	2022	3D-Mask RCNN	3.0 T/ SAG PDW	Detection of meniscal tear and normal meniscus	0.907
Shin et al. [48]	2022	CNN	1.5 T/ FS T2 W COR SAG	Detection of MM LM tear, type of the lesion: complex, radial, horizontal, longitudinal	MM 0.889 LM 0.817 BM 0.924
Bien et al. [12]	2018	CNN, MRNET	1.5 T/3 T/ SAG T2 W COR T1 W Axial PDW	Detection of meniscal tears, ACL tears, and abnormalities	0.847
Pedoia et al. [13]	2019	3D CNN, 2D U-NET	3 T/ 3D FSE CUBE	Detection of meniscal tears, ACL tears, and cartilage lesions	0.89
Tsai et al. [49]	2020	CNN ELNET	1.5 T/3 T/ SAG T2 W COR T1 W Axial PDW	Meniscal tear detection	0.904
Astuto et al. [50]	2021	3D CNN	3 T/ 3DFSE CUBE	Anterior and posterior horn MM, anterior and posterior LM	0.93

**Table 3 diagnostics-14-01090-t003:** Approaches in each step of the DL pipeline.

Authors	Initial Dataset	Preprocessing	Localization ROI	Ground Truth	Curated Dataset
Couteaux et al. [41]	1128 images (2D)	Morphological preprocessing approximate menisci	Both menisci Identify tears in each meniscus with Mask R-CNN	Batch 1 contained 55/257 (21.4%) images with horizontal posterior tears, 46/257 (17.9%) with vertical posterior tears, 13/257 (5.1%) with horizontal anterior tears, and 8/257 (3.1%) with vertical anterior tears. Batch 2 contained 107/871 (12.3%) images with horizontal posterior tears, 60/871 (6.9%) with vertical posterior tears, 8/871 (0.9%) with horizontal anterior tears, and 3/871 (0.3%) with vertical anterior tears	1128 images
Roblot et al. [42]	1123 images (2D)	Bounding box surrounding each horn of the meniscus Labeling	Detection technique Fast RCNN and Faster RCNN	Annotation data in a CSV file 1948 normal 248 meniscal tears 183 horizontal tears 115 vertical tears	2246 images
Fritz et al. [43]	20,520 images, 100 patients (2D)	Automated selection of coronal and sagittal images by DCNN Cropping around the meniscus	Class activation map (CAM) of the last convolution layer in the CNN and maps to an axial knee image.	The ground truths (binary labels) used to train the CNN were extracted from human-produced, anonymized clinical reports belonging to the MRI studies using rule-based natural language processing (NLP) algorithm.	18,520 images
Rizk et al. [45]	11,353 Images, 10,401 patients (2D)	The content was extracted using natural language processing (NLP) algorithms. Manually annotated by two data scientists 3D bounding boxes normalized in the range [0, 1] around medial and lateral menisci	Custom localization In-house annotation tool	8058 training 299 testing Five musculoskeletal specialists were compared to general radiologists’ reports. External validation was performed using the publicly available MRNet database	8058 images
Tack et al. [44]	2399 scans (3D)	Resampling and resizing cropped data automated preprocessing step Bounding box regression task	Bonding box crop approach for segmentation of both menisci	WORMS scores	2399 scans
Li et al. [46]	924 Patients, 18 images/ patient (2D)	Data augmentation	Mask R–CNN	The total number of labels in the training, validation, and internal testing dataset was 30,080.	30,080 images
Li Yz et al. [47]	533 Patients	Labels of meniscal tears were based on arthroscopic reports from the same group of orthopedic surgeons as a standard of reference	ROI using ITK-SNAP on the sagittal section. The meniscus was manually segmented A 3D-mask RCNN for the detection and segmentation of meniscus	Two radiologists and software based on DCNN retrospectively evaluated clinical patients’ knee MRIs to detect medial and lateral meniscus tears	533 images
Shin et al. [48]	599 cases (3D)	Assessed by two board-certified orthopedic knee specialists, intra-class correlation coefficients	n/a	268 horizontal tears 147 complex tears 48 radial tears, 75 longitudinal tears 449 cases without meniscal tears	1048 cases
Bien et al. [12]	1370 images (17–61 slices) (2D)	Manually reviewed to curate a dataset. Histogram-based intensity standardization	Reference standard labels by internal validation. Vote of 3 practicing board-certified MSK radiologists	1104 abnormal 319 ACL 508 MT 194 ACL+MT External validation of 917 images	1370 images
Pedoia et al. [13]	1481 images (3D)	Annotated by five board-certified radiologists	2D-Unet 3D bounding box	WORMS scores	1481 images
Tsai et al. [49]	1370 images (17–61 slices) (2D)	Histogram-based intensity standardization Randomized data augmentations to each series	n/a	1104 abnormal 319 ACL 508 MT 194 ACL+MT External validation of 917 images	1370 images
Astuto et al. [50]	1435 Images, 294 patients (3D)	Five board-certified radiologists graded no overlapping portions of the dataset	VNet 11 regions of interest (ROIs)	WORMS-based inference	1435 images

## Data Availability

The original contributions presented in the study are included in the article, further inquiries can be directed to the corresponding authors.

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
