# Peer review of "A Comprehensive Evaluation of Deep Learning Models on Knee MRIs for the Diagnosis and Classification of Meniscal Tears: A Systematic Review and Meta-Analysis"

_diagnostics, 2024, doi:10.3390/diagnostics14111090_

Round 1

Reviewer 1 Report (Previous Reviewer 1)

Comments and Suggestions for Authors

Even the authors have significantly revised the paper, there are still some concerns and comments need to be addressed.

In particular, it is not clear the contribution of each research study on the description of the evaluation applied to DL models. The explanation of the ROI localization is not sufficiently decsribed. Also, it is not clearly reported the most significant methods for image preprocessing and their connection with the review studies.

The limitations of the study are poorly defined and do not provide an efficient explanation for the reader.

Overall, it is well organized.

Comments on the Quality of English Language

some English errors

Author Response

Response to Reviewer 1

We greatly appreciate your valuable comments and recommendations. They have been thoroughly considered and incorporated into our revisions.

Reviewer 1: Even the authors have significantly revised the paper, there are still some concerns and comments need to be addressed.

In particular, it is not clear the contribution of each research study on the description of the evaluation applied to DL models. The explanation of the ROI localization is not sufficiently decsribed. Also, it is not clearly reported the most significant methods for image preprocessing and their connection with the review studies.

Figure 8 and Table 2 depict each study's contribution to the description and evaluation of each DL model according to the CLAIM checklist and the risk of bias obtained with the QUADAS-2 tool.

We have included the following section to clarify the most significant methods for image preprocessing and their connection with the review studies and the explanation of the ROI localization:

Description of the image preprocessing

To obtain the curated dataset ready to fit into the model, all the teams, without exception, had intensely pre-processed the original knee MRIs. Table 4 summarizes the pre-processing techniques applied in each study. The most important preprocessing techniques applied in the studies are:

  • Resizing and resampling: The images varied in resolutions and dimensions. Resizing and resampling the images to a consistent size or spacing was essential for ensuring compatibility with deep-learning models requiring fixed input dimensions. Dimensionality reduction was achieved by manual cropping [45] and bounding boxes [44] or by deep models such as UNet [13].
  • Intensity normalization and histogram equalization: The MRIs exhibited varying intensity ranges and contrasts due to differences in acquisition protocols and devices. Intensity normalization scaled the pixel values to a standard range, such as [0, 1] or [0, 255], enhancing the contrast and enabling more meaningful image comparisons. A histogram-based intensity standardization algorithm was applied to the images to account for variable pixel intensity scales within the MRI series [12, 49]. A representative intensity distribution was learned from the training set exams for each series. Then, the parameters of this distribution were used to adjust the pixel intensities of exams in all datasets (training, tuning, and validation). Under this transformation, pixels with similar values correspond to similar tissue types. After intensity standardization, pixel values were clipped between 0 and 255, the standard range for PNG images [12, 49].
  • Image noise and signal abnormalities reduction: MRIs can be affected by various types of noise, such as Gaussian noise, salt-and-pepper noise, or speckle noise. Noise reduction techniques, such as Gaussian filtering, median filtering, or anisotropic diffusion, reduce noise while preserving important image features. The model's performance is very much dependent on the image quality of the training dataset. One profound influence on the model performance is the image nosiness or the artifacts [42]. Signal abnormalities in images are still a challenge [44]. When menisci with tears are to be distinguished from menisci without tears, signal abnormalities are currently regarded as the latter. A fine-grained differentiation between tears and signal abnormalities is a challenge, primarily because of the ambiguous image appearance [44].
  • Data augmentation: Data augmentation was used to increase the size and diversity of a training dataset by creating new instances by applying various transformations to the original data. Most studies have imbalanced sets for the categories to be used for training. Usually, there are fewer images of healthy meniscus than of torn ones. Also, as in the case of the predefined datasets [41, 42], those contained only two T2-weighted MR images in the sagittal plane for each patient, whereas MRI examination of the knee usually includes around 100 images. Moreover, images were pre-processed to have the same matrix and voxel size. Dataset imbalance may explain the inferior overall performances on lateral meniscal tear detection and characterization. A more significant amount of data, including lateral meniscal tears in the training dataset, may further increase model performances laterally [45]. Tsai et al. [49] performed randomized data augmentations to each series, which included translation, horizontal scaling, and minor rotations up to 10 degrees around the center of the volume. For volumes captured in the axial and coronal orientations, they applied an additional random rotation of a multiple of 90 degrees to the volume. Finally, all the images were resized to 256 x 256 before entering the network. Aside from data augmentation, they implemented oversampling to compensate for dataset imbalance [49].

Description of the ROI localization

Accurate segmentation of regions of interest (ROIs), in our case, the menisci, is one of the most important steps for the DL model performance. However, segmentation can be challenging due to overlapping structures, weak boundaries, or similar intensities between the target region and surrounding tissue. Segmentation techniques ranged from simple thresholding to more complex approaches like active contours or deep learning-based methods. Careful selection and application of segmentation techniques have significantly improved the input images' quality and enhanced subsequent analysis algorithms' performance. Table 4 summarizes various techniques applied to the identification of ROIs, such as custom localization with manual annotation tools, class activation maps, bounding box crop approach, semi-automated crops using ITK-SNAP, segmentation with DL models UNet, fast–CNN, bounding box regression, and 2D and 3D bounding boxes.

Most of the presented methods individually detected meniscal tears for all anatomical sub-regions of the menisci, i.e., the anterior horn, the meniscal body, and the posterior horn.”

Reviewer 1: The study's limitations are poorly defined and do not provide an efficient explanation for the reader.

While this review and meta-analysis offer valuable insights into how deep learning (DL) contributes to identifying meniscal tears on MRI scans, there are notable limitations to consider. Firstly, the studies included vary in terms of the datasets used, MRI techniques employed, DL models utilized, and the absence of clear standard references, potentially leading to result and interpretation discrepancies. Moreover, only a few studies compare DL outcomes with arthroscopic surgery, limiting the applicability of findings to clinical contexts. Arthroscopy is considered the gold standard for non-invasive knee MRI assessments due to its ability to directly evaluate structural integrity, tear patterns, and functional characteristics. Additional studies like those by Ying et al. [51] are necessary to integrate arthroscopic insights into MRI diagnostic models for more precise meniscus tear detection. Furthermore, despite the focus on DL model performance regarding meniscal tear localization, description, or classification, the number of studies addressing these aspects remains relatively small, potentially restricting the depth of analysis. Although several similar reviews[30-32, 52] exist in the literature on this topic, the primary limitations remain the heterogeneity and number of included studies, preventing thorough statistical comparisons of output modes. Addressing these limitations in future research endeavors would enhance the comprehensive evaluation of DL's role in detecting meniscal tears on MRI scans.

Reviewer 1: Overall, it is well organized.

some English errors

English proofreading has been applied.

Reviewer 2 Report (New Reviewer)

Comments and Suggestions for Authors

The paper presents a review of the deep learning methods for knee MR image analysis. It specifically focuses on the detection of meniscal tears in MRI images. The authors put a good effort into gathering the data, however, there are a few suggestions by the reviewer that can help improve the paper's contents and also its presentation.

In the abstract of the manuscript, it is claimed that the DL-based methods for meniscal tears detection surpass the expertise of musculoskeletal radiologists. This is a very strong statement and must be justified with extensive experimental evaluation, quoting a single reference is not enough. 

The literature discussed in the study does not cover recent research on this subject. For example, there are only three papers from 2022 and only one paper from 2023 have been included in the study.

The literature covered in the study is insufficient and outdated. There are around 40 papers covered in the study. The authors should include more papers to present a wider view of the research done on this topic.

No doubt that DL-based methods are performing well in segmentation and classification topics, the feature-based machine learning is still a useful tool in many applications. These methods should also be included in the study and a contrast between the DL and ML should be presented. I am sure it would be of great interest to the readers of the paper.

Calculating the sensitivity, specificity, etc metrics is fundamental, the reference [38] can be dropped.

The presentation of the paper needs some serious work. Most figures presented in the manuscript are of poor print quality, e.g., Fig. 11, 12, and 13. Moreover, it is not clear if the authors ran those experiments and computed these results or borrowed these figures from the references. The choice of dataset is also not clear.

A minor comment: Please use the journal's proper template for preparing your manuscript.

Comments on the Quality of English Language

NA

Author Response

Response to Reviewer 2

We greatly appreciate your valuable comments and recommendations. They have been thoroughly considered and incorporated into our revisions.

Reviewer 2: The paper presents a review of the deep learning methods for knee MR image analysis. It specifically focuses on the detection of meniscal tears in MRI images. The authors put a good effort into gathering the data, however, there are a few suggestions by the reviewer that can help improve the paper's contents and also its presentation.

Reviewer 2: In the abstract of the manuscript, it is claimed that the DL-based methods for meniscal tears detection surpass the expertise of musculoskeletal radiologists. This is a very strong statement and must be justified with extensive experimental evaluation, quoting a single reference is not enough.

We have changed the text as follows:

The study's findings underscore that a range of deep-learning models exhibit robust performance in detecting and classifying meniscal tears, in one case even surpassing the expertise of musculoskeletal radiologists.

Reviewer 2: The literature discussed in the study does not cover recent research on this subject. For example, there are only three papers from 2022 and only one paper from 2023 have been included in the study.

We started to write this review in September 2023. Therefore, the initial search was stopped in September 2023. What studies were allowed (DL model, finding and diagnosing meniscal tears only or along with ACL tears) and what studies were not allowed (studies that did not use the DL model, studies that did not use MRI images, studies that only looked at other injuries like ACL tears, studies that looked at segmentation of the meniscus). The search resulted in twelve studies that met our criteria. However, during the evaluation process of our review, we have included other two studies that match our criteria, published after the initial search (one in 2023, the other from 2024).

This paper focuses exclusively on deep learning techniques and only on the menisci. We wanted to analyze the state of the art to see the challenges and limitations of this technique for future improvements. Machine learning solves problems through statistics and mathematics and manually selects and extracts features from raw data, while deep learning combines statistics and mathematics with neural network architecture. For more clarity, we have included the following text in the review:

The search and inclusion of articles into this systematic review have been narrowed to only deep learning models applied to diagnosing menisci in magnetic resonance imaging. The reason for such an approach is manifold. Firstly, due to the increasing popularity of deep learning models, especially DCNNs, we intended to analyze the state of the art and the challenges and limitations of the variety of such models applied to investigating menisci diseases. As previously mentioned, the pathology of the menisci is a frequent issue worldwide and in all age groups. On the other hand, MRI is one of the most common methods of investigation. Deep learning models' structure and operation differ from other machine learning techniques (e.g., support vector machines, decision trees, random forests, and logistic regression). These ML methods typically involve manually selecting and extracting features from images, which are then used for prediction or classification. Unlike traditional ML methods, DL models can automatically learn and extract hierarchical features from raw data, speeding up the process. Thus, DL techniques and network-based computation have been widely adopted in various domains, including medical imaging. Consequently, we focused on the potential of DL models in diagnosing menisci in MRIs, excluding other machine learning techniques, in the scope of the homogeneity of articles that can be further quantitatively and qualitatively assessed. We have evaluated articles that present deep learning techniques and their potential to further advance a new avenue for diagnosis and treatment strategies. This paper details these methods, results, and the implications of such findings for future research.”

 Reviewer 2: The literature covered in the study is insufficient and outdated. There are around 40 papers covered in the study. The authors should include more papers to present a wider view of the research done on this topic.

When one searches PubMed or Google Scholar, many published articles relate to artificial intelligence and meniscus injuries. However, when applying the inclusion and exclusion criteria set for this review, only 12 articles remain relevant to its scope. As of May 1st, 2024, a new search that met our criteria had supplemented the review with two papers.

Reviewer 2: There is no doubt that DL-based methods are performing well in segmentation and classification topics, the feature-based machine learning is still a useful tool in many applications. These methods should also be included in the study and a contrast between the DL and ML should be presented. I am sure it would be of great interest to the readers of the paper.

We concentrated solely on evaluating the capability of a deep learning model for diagnosing meniscus tears. We chose this focus because previous reviews (e.g., Kunze et al., Siouras et al., and Santomartino et al.) have already discussed both deep learning and machine learning methods. Furthermore, our review solely focuses on deep learning models due to the established superiority of deep learning over machine learning in knee MRI image analysis.

Reviewer 2: Calculating the sensitivity, specificity, etc metrics is fundamental, the reference [38] can be dropped.

We have eliminated reference 38.

Reviewer 2: The presentation of the paper needs some serious work. Most figures presented in the manuscript are of poor print quality, e.g., Fig. 11, 12, and 13. Moreover, it is not clear if the authors ran those experiments and computed these results or borrowed these figures from the references. The choice of dataset is also not clear.

We have improved the printing quality of Figures 11, 12, and 13. The figures are the results of the experiments we computed using Meta-DiSc 1.4 software.

The Cochrane guidelines for inter-study variation guided the selection of the dataset. We fitted a random-effects model for estimation with partial pooling because methodological and interpretation differences made the true effect size for all studies unlikely to be identical. We conducted subgroup analyses to investigate the sources of heterogeneity. We gave summary estimates for the prediction accuracy of AI models once we minimized heterogeneity and removed outliers. We performed all data analysis and visualization using the Meta-DiSC tool [39].

A minor comment: Please use the journal's proper template for preparing your manuscript.

For this version, we used the journal’s template. English proofreading has been applied.

Reviewer 3 Report (New Reviewer)

Comments and Suggestions for Authors

The manuscript discussed and evaluated effectiveness of deep learning models in recognizing, localizing, describing, and categorizing meniscal tears in MRI images. A systematic analysis of carefully selected work from the literature was presented. Conclusions follow a detailed and critical analysis. The presented work would be of value and support focused research in related areas. 

Author Response

Thank you very much for your review.

Round 2

Reviewer 2 Report (New Reviewer)

Comments and Suggestions for Authors

I appreciate the authors for improving the paper. The paper's contribution is very limited and only a few techniques from the literature are selected based on criteria designed by the authors. They should revisit it. Critical analysis of the techniques is missing. It's a simple narration of what is described in the related work. Therefore, my opinion and the previous rating of this manuscript are unchanged.

Comments on the Quality of English Language

The paper's contribution is very limited and only a few techniques from the literature are selected based on criteria designed by the authors. They should revisit it. Critical analysis of the techniques is missing. It's a simple narration of what is described in the related work. Therefore, my opinion and the previous rating of this manuscript are unchanged.

Author Response

I appreciate the authors for improving the paper. The paper's contribution is very limited and only a few techniques from the literature are selected based on criteria designed by the authors. They should revisit it. Critical analysis of the techniques is missing. It's a simple narration of what is described in the related work. Therefore, my opinion and the previous rating of this manuscript are unchanged.

By focusing solely on DL models, we have presented a comprehensive overview (meta-analyses, statistics, and narratives) of such models' accomplishments, challenges, and limitations in investigating and diagnosing meniscus pathology. We also outlined future directions for improvement and progress for DL models in the vast MRI field, using knee and meniscus pathology as examples. We believe such specific and focused studies can effectively highlight the limitations and stimulate further advancements.

Besides, in 2024, deep learning will excel in complex pattern recognition and achieve new heights in efficiency, leading to cutting-edge AI solutions in many domains, including medical imaging. Looking ahead, DL will continue to evolve rapidly. Therefore, your recommendation to include an assessment of the ML techniques is beyond the scope of this study and would largely extend its length.  Also, it was not our scope to study ML techniques other than DL models because ML's strength lies in its simplicity and interpretability, making it ideal for smaller, structured datasets and problems. At the same time, DL excels in handling complex, unstructured data, offering unparalleled performance in computer vision and natural language processing tasks. The two techniques differ from many points of view, leading to the impossibility of thorough comparisons and meta-analyses.

Critical analysis of the techniques can be directly depicted from the CLAIM checklist evaluation and the risk of bias, as well as from the discussion and conclusions section. However, in this new manuscript version, we have revised the narrative within the results section to be concise and straightforward. We have also strengthened the conclusions by summarizing the most important research directions that can be drawn from our review. All changes are highlighted in blue.

Finally, we consider that comprehensively analyzing twelve different DL models and their preprocessing techniques is not trivial and brings value and direction for further research.  For example, another similar study published in Feb 2024 in the European Radiology Journal (Springer, IF 5.9) included eleven studies. (see Zhao Y, Coppola A, Karamchandani U, Amiras D, and Gupte CM.). Artificial intelligence applied to magnetic resonance imaging reliably detects the presence, but not the location, of meniscus tears: a systematic review and meta-analysis. Eur Radiol. 2024 Feb 22. doi: 10.1007/s00330-024-10625-7. Epub ahead of print. PMID: 38386028.)

This manuscript is a resubmission of an earlier submission. The following is a list of the peer review reports and author responses from that submission.

Round 1

Reviewer 1 Report

Comments and Suggestions for Authors

There are some major concerns to be considered in the revised version.

Definition of Review Gap:

The review article fails to clearly define the review gap, leaving the reader without a distinct understanding of why this review contributes significantly to the existing body of literature. To strengthen the manuscript, the authors should explicitly highlight how their work advances beyond previous review studies in the field and articulate the unique contributions this review brings to the domain of deep learning applications on MRI Knee Scans.

 Insufficient Result Comparison:

The review lacks a detailed and comprehensive comparison of results presented in the various reviewed studies. A more thorough analysis and synthesis of findings across different works are needed to provide a nuanced understanding of the trends, patterns, and inconsistencies in the applications of deep learning in the domain of MRI knee diagnosis and Classification of Meniscal Tears.

 Graphical Representation of Results:

The paper should incorporate graphical representations to facilitate a clearer understanding of the comparative results. Visual aids such as charts or graphs could enhance the presentation of findings and contribute to the overall clarity of the review.

 Increase Citations for a Comprehensive Review:

The number of cited works in the review appears to be limited. To strengthen the academic rigor and comprehensiveness of the study, the authors should consider including a broader range of relevant literature. This will ensure a more robust foundation for the review and provide readers with a broader perspective on the subject.

Clarification and Unification of Metrics:

The article should offer a more detailed explanation of the metrics used in the reviewed studies. Additionally, if possible, the authors should consider unifying the metrics to facilitate a more meaningful and standardized comparison across different applications of ML/DL in MRI Knee diagnosis and Classification of Meniscal Tears.

Addressing these points will significantly enhance the overall quality and impact of the review article, providing a more valuable resource for researchers and practitioners in the field.

Comments on the Quality of English Language

Minor comments

Author Response

Comments and Suggestions for Authors

There are some major concerns to be considered in the revised version.

Definition of Review Gap:

The review article fails to clearly define the review gap, leaving the reader without a distinct understanding of why this review contributes significantly to the existing body of literature. To strengthen the manuscript, the authors should explicitly highlight how their work advances beyond previous review studies in the field and articulate the unique contributions this review brings to the domain of deep learning applications on MRI Knee Scans.

In the introduction, we emphasized the uniqueness of our review compared to others, outlined the study's objectives, and incorporated a meta-analysis as well.

 Insufficient Result Comparison:

The review lacks a detailed and comprehensive comparison of results presented in the various reviewed studies. A more thorough analysis and synthesis of findings across different works are needed to provide a nuanced understanding of the trends, patterns, and inconsistencies in the applications of deep learning in the domain of MRI knee diagnosis and Classification of Meniscal Tears.

We have significantly enhanced our findings by incorporating a comparison of datasets, detailing the acquisition protocol, refining the classification of meniscal tears, conducting a meta-analysis, and including an assessment of bias risk.

 Graphical Representation of Results:

The paper should incorporate graphical representations to facilitate a clearer understanding of the comparative results. Visual aids such as charts or graphs could enhance the presentation of findings and contribute to the overall clarity of the review.

More charts , tables, graphs and diagrams of our work had been added.

 Increase Citations for a Comprehensive Review:

The number of cited works in the review appears to be limited. To strengthen the academic rigor and comprehensiveness of the study, the authors should consider including a broader range of relevant literature. This will ensure a more robust foundation for the review and provide readers with a broader perspective on the subject.

 The number of the cited works was doubled.

Clarification and Unification of Metrics:

The article should offer a more detailed explanation of the metrics used in the reviewed studies. Additionally, if possible, the authors should consider unifying the metrics to facilitate a more meaningful and standardized comparison across different applications of ML/DL in MRI Knee diagnosis and Classification of Meniscal Tears.

We have also elucidated the metrics to ensure the manuscript is easily comprehensible for reviewers and other practitioners.

Addressing these points will significantly enhance the overall quality and impact of the review article, providing a more valuable resource for researchers and practitioners in the field.

Thank you for your suggestion. We trust that we have considered all the details provided, resulting in a significant improvement in our work.

Reviewer 2 Report

Comments and Suggestions for Authors

The manuscript provides an extensive analysis of deep learning models in the context of MRI knee scans for meniscal tear diagnosis. The authors systematically review various studies, focusing on the effectiveness of deep learning models compared to traditional diagnostic methods. My main concern is the lack of novelty in the present study since we have similar papers with stronger methodologies:
1- https://link.springer.com/article/10.1007/s00256-023-04416-2
2- https://www.sciencedirect.com/science/article/abs/pii/S0749806320307441
3- https://www.mdpi.com/2075-4418/12/2/537

Author Response

Comments and Suggestions for Authors

The manuscript provides an extensive analysis of deep learning models in the context of MRI knee scans for meniscal tear diagnosis. The authors systematically review various studies, focusing on the effectiveness of deep learning models compared to traditional diagnostic methods. My main concern is the lack of novelty in the present study since we have similar papers with stronger methodologies:
1- https://link.springer.com/article/10.1007/s00256-023-04416-2
2- https://www.sciencedirect.com/science/article/abs/pii/S0749806320307441
3- https://www.mdpi.com/2075-4418/12/2/537

Thank you for your review , we improved our research as follows -Our systematic review introduces novel approaches compared to similar previous studies: i). focus is given exclusively on meniscal tears, and while previous reviews have explored DL models in conjunction with various knee injuries, none have explicitly outlined meniscal lesion classifications [1,2,3]; ii). the accuracy of the DL model is studied through meta-analyses that previous reviews have not conducted to the best of our knowledge [1,2,3]; iii). evaluates whether the presented DL models outperform radiologist interpretations; iv.) evaluates the DL model's performance in classifying meniscal tears and detecting their various types, providing an updated comparison with human interpretation and, most notably, v.) introduces the novelty of assessing DL model performance against arthroscopic surgery as the standard reference by the end of 2023

Reviewer 3 Report

Comments and Suggestions for Authors

the manuscript provides a comprehensive overview of the application of deep learning techniques, specifically convolutional neural networks (CNNs), in the detection and classification of meniscal tears in knee MRI images. The systematic review follows the PRISMA guidelines, and the study's findings suggest that deep learning models exhibit robust performance, sometimes surpassing the expertise of musculoskeletal radiologists. The manuscript covers various aspects of deep learning applications in this domain, including challenges, data preparation, CNNs, model training, validation, testing, and clinical application.

Here are some comments and suggestions for improvement:

  • It might be helpful to provide a brief definition or explanation of meniscal tears and their clinical significance at the beginning of the introduction for readers who may not be familiar with the topic.
  •  
  • The introduction could benefit from a clearer statement of the research question or hypothesis to guide the reader.
  •  
  • Elaborate more on the challenges faced in obtaining a diverse and representative dataset for training deep learning models.
  •  
  • Expand on the challenges faced in setting up the best parameters for CNNs. Provide more details on the considerations for fine-tuning and running iterations.
    Clarify the role of transfer learning in improving models. Explain how researchers applied transfer learning in the context of meniscal tear detection.
    Discuss any regulatory or ethical considerations associated with deploying deep learning models in clinical settings. Address any potential limitations or risks.
    Provide concrete examples or evidence supporting statements on consistency, efficiency, and early detection to strengthen the argument.
    The conclusion could summarize the key findings more explicitly and highlight the implications of the study for future research or clinical practice.
    Discuss the limitations of the reviewed studies in terms of sample size, study design, and potential biases.
    Consider adding a section on potential future research directions or areas that need further exploration in the field of deep learning for meniscal tear detection.
Comments on the Quality of English Language

Add transitional sentences or phrases between paragraphs to create a smoother flow between different sections of the manuscript.

In a few places, the use of more precise language could enhance clarity. For example, consider replacing phrases like "accorded thorough searches" with more straightforward language.

Overall, the language quality is quite good, and the suggested improvements are minor. A careful review with attention to sentence structure, word choice, and consistency will further enhance the manuscript's clarity and readability.

Author Response

Comments and Suggestions for Authors

the manuscript provides a comprehensive overview of the application of deep learning techniques, specifically convolutional neural networks (CNNs), in the detection and classification of meniscal tears in knee MRI images. The systematic review follows the PRISMA guidelines, and the study's findings suggest that deep learning models exhibit robust performance, sometimes surpassing the expertise of musculoskeletal radiologists. The manuscript covers various aspects of deep learning applications in this domain, including challenges, data preparation, CNNs, model training, validation, testing, and clinical application.

Here are some comments and suggestions for improvement:

  • It might be helpful to provide a brief definition or explanation of meniscal tears and their clinical significance at the beginning of the introduction for readers who may not be familiar with the topic.

  • The introduction could benefit from a clearer statement of the research question or hypothesis to guide the reader.
  •  
  • Elaborate more on the challenges faced in obtaining a diverse and representative dataset for training deep learning models.
  •  

Expand on the challenges faced in setting up the best parameters for CNNs. Provide more details on the considerations for fine-tuning and running iterations.

Clarify the role of transfer learning in improving models. Explain how researchers applied transfer learning in the context of meniscal tear detection.

Discuss any regulatory or ethical considerations associated with deploying deep learning models in clinical settings. Address any potential limitations or risks.

Provide concrete examples or evidence supporting statements on consistency, efficiency, and early detection to strengthen the argument.

The conclusion could summarize the key findings more explicitly and highlight the implications of the study for future research or clinical practice.

Discuss the limitations of the reviewed studies in terms of sample size, study design, and potential biases.

Consider adding a section on potential future research directions or areas that need further exploration in the field of deep learning for meniscal tear detection.

Thank you for your valuable suggestions and recommendations. As a result, we have made several enhancements to our manuscript. These include providing a concise explanation of meniscal tears in the introduction, clarifying our methods, offering detailed descriptions of the DL Models, assessing the risk of bias and heterogeneity in the studies. Additionally, we have discussed the limitations of our review and the studies included, and we have strengthened the conclusion by emphasizing avenues for future research.

Comments on the Quality of English Language

Add transitional sentences or phrases between paragraphs to create a smoother flow between different sections of the manuscript.

In a few places, the use of more precise language could enhance clarity. For example, consider replacing phrases like "accorded thorough searches" with more straightforward language.

Overall, the language quality is quite good, and the suggested improvements are minor. A careful review with attention to sentence structure, word choice, and consistency will further enhance the manuscript's clarity and readability.